# Lipoxygenase Pathways in Diatoms: Occurrence and Correlation with Grazer Toxicity in Four Benthic Species

**DOI:** 10.3390/md18010066

**Published:** 2020-01-19

**Authors:** Nadia Ruocco, Genoveffa Nuzzo, Giuliana d’Ippolito, Emiliano Manzo, Angela Sardo, Adrianna Ianora, Giovanna Romano, Antonella Iuliano, Valerio Zupo, Maria Costantini, Angelo Fontana

**Affiliations:** 1Istituto di Chimica Biomolecolare, Consiglio Nazionale delle Ricerche, Via Campi Flegrei 34, 80078 Pozzuoli, Napoli, Italy; nadia.ruocco@gmail.com (N.R.); nuzzo.genoveffa@icb.cnr.it (G.N.); gdippolito@icb.cnr.it (G.d.); emanzo@icb.cnr.it (E.M.); angela.sardo@szn.it (A.S.); 2Department of Marine Biotechnology, Stazione Zoologica Anton Dohrn, Villa Comunale, 80121 Napoli, Italy; ianora@szn.it (A.I.); valerio.zupo@szn.it (V.Z.); mcosta@szn.it (M.C.); 3Istituto per le Applicazioni del Calcolo “Mauro Picone”, Consiglio Nazionale delle Ricerche, Via Pietro Castellino 111, 80131 Napoli, Italy; aiuliano19@gmail.com

**Keywords:** chemical ecology, mass spectrometry, diatoms, oxylipins, lipoxygenase, biosynthesis, fatty acid derivatives

## Abstract

Oxygenated derivatives of fatty acids, collectively called oxylipins, are a highly diverse family of lipoxygenase (LOX) products well described in planktonic diatoms. Here we report the first investigation of these molecules in four benthic diatoms, *Cylindrotheca closterium*, *Nanofrustulum shiloi*, *Cocconeis scutellum*, and *Diploneis* sp. isolated from the leaves of the seagrass *Posidonia oceanica* from the Gulf of Naples. Analysis by hyphenated MS techniques revealed that *C. closterium*, *N. shiloi*, and *C. scutellum* produce several polyunsaturated aldehydes (PUAs) and linear oxygenated fatty acids (LOFAs) related to the products of LOX pathways in planktonic species. *Diploneis* sp. also produced other unidentified fatty acid derivatives that are not related to LOX metabolism. The levels and composition of oxylipins in the benthic species match their negative effects on the reproductive success in the sea urchin *Paracentrotus lividus*. In agreement with this correlation, the most toxic species *N. shiloi* revealed the same LOX pathways of *Skeletonema marinoi* and *Thalassiosira rotula*, two bloom-forming planktonic diatoms that affect copepod reproduction. Overall, our data highlight for the first time a major role of oxylipins, namely LOFAs, as info-chemicals for benthic diatoms, and open new perspectives in the study of the structuring of benthic communities.

## 1. Introduction

Diatoms are primary producers in marine environments and are traditionally considered a good source of food for both planktonic and benthic consumers [1,2]. The beneficial role of diatoms in marine food webs has however been questioned with the first reports demonstrating that the reproductive success of the main group of planktonic grazers, crustacean copepods, is impaired by heavy ingestion of diatoms during marine blooms [3,4,5]. Firstly, abortions and congenital malformations observed in copepods were correlated with the lipoxygenase (LOX) production of polyunsaturated aldehydes (PUAs) in diatoms [6]. These metabolites, whose chemistry was later revised in decatrienal, octadienal, octatrienal, and heptadienal [7,8,9,10], were proposed to be part of a chemical defence mechanism of planktonic diatoms against herbivorous zooplankton [11,12,13,14].

In analogy to the synthesis of hexenal and nonenal in terrestrial plants [15,16], diatom PUAs derive from oxidation of membrane-derived polyunsatured fatty acids (PUFAs) by the lipoxygenase (LOX)/hydroperoxide lyase (HPL) pathway [17,18,19,20]. However, PUAs occur only in a small fraction of diatom species [21,22], whereas almost all diatoms possess oxidative pathways for the synthesis of linear oxygenated fatty acid derivatives (LOFAs) that preserve the alkyl chain of the original fatty acid precursors and incorporate hydroxy-, keto-, oxo- and epoxy functionalities.

The chemistry of lipoxygenase products, collectively named oxylipins, was extensively studied in planktonic diatoms [8,9,23,24,25,26] but there are only a few reports on PUAs for benthic species including *C. scutellum parva* [27], *Achnanthes biasolettiana* [28], *Tabularia affinis*, *Proschkinia complanatoides* and *Navicula* sp. [29]. Furthermore, other studies showed a reduction in hatching success and grazing of macroinvertebrates in response to organic extracts of benthic diatoms [28,30]. Recently, we reported that feeding on four benthic species, namely *Cylindrotheca closterium*, *Nanofrustulum shiloi*, *Diploneis* sp. and *Cocconeis scutellum*, affected reproduction in the sea urchin *Paracentrotus lividus* [31,32]. 

This work is aimed at the characterization of oxylipins and lipoxygenase pathways in these four benthic diatoms in order to correlate LOX products and LOX activity to the detrimental effect on marine grazers. Protocols were adapted from methods previously applied to planktonic species [26,33,34].

## 2. Results

### 2.1. LC-MS Analysis of LOFAs

Monoclonal cultures of the four benthic species were maintained in *f*/*2* medium (Sigma Guillard’s). Massive cultures were prepared and collected by centrifugation (for more details, see Section 4.2), then frozen in liquid nitrogen and kept at −80 °C until analysis. Oxylipins were extracted using the method reported in references [26,34].

Identification and intraspecific distribution was supported by LC-MS analysis on the microalgal extracts of the four benthic diatoms [8,26,33,34,35] (Figure 1). In analogy with planktonic species [23,25,36,37,38,39], results on *C. closterium*, *N. shiloi* and *C. scutellum* revealed characteristic presence of hydroxy fatty acids (*m*/*z* 355, λ_max_ = 234 nm, 11-HEPE and 15-HEPE) and epoxyalcohol fatty acids (*m*/*z* 371, 14,13-EHETE) of eicosa-5*Z*,8*Z*,11*Z*,14*Z*,17*Z*-pentaenoic acid (EPA, C20:5 ω3) even if the three species displayed different composition and levels of these compounds (Figure 1). 

In *C. closterium*, we also detected the presence of an epoxyalcohol derivative (*m*/*z* 397, 16,15-EHDPE) of docosa-4*Z*,7*Z*,10*Z*,13*Z*,16*Z*,19*Z*-esenoic acid (DHA, C22:6 ω3) [38] while *N. shiloi* showed derivatives of C_16_ fatty acids including the canonical epoxyalcohol (*m*/*z* 319, 11,10-EHETE) of 6*Z*,9*Z*,12*Z*- hexadecatrienoic acid (HTrA, C16:3 ω4) together with two oxygenated derivatives of palmitoleic acid (PALOL, C16:1), namely 9-hydroxyhexadec-7*E*-enoic acid (*m*/*z* 307 [M + Na]^+^, 9-HHME, RT= 11.27, λ_max_ = 214 nm, 0.012 ± 0.0009 μmol mg^−1^ C) and 9-ketohexadec-7*E*-enoic acid (*m*/*z* 305 [M + Na]^+^, 9-KHME, RT= 10.20, 0.0014 ± 0.0001 μmol mg^−1^ C) [25].

Finally, the LC-MS profile of *Diploneis* sp. (Figure 1d) revealed two pairs of isobaric metabolites with molecular weight (M + Na^+^) at *m*/*z* 381 and 345. 

The spectrometric data of these compounds were not consistent with the structures of known oxylipins even if their chemical signature is in agreement with oxygenated derivatives of fatty acids. According to these data, statistical analysis showed significantly different levels of oxylipins in the four species (*p*-value < 0.001). In particular, *N. shiloi* was found to contain the highest level of LOX products among the four species, followed by *C. closterium* and *C. scutellum* (Figure 2a). As no canonical oxylipins were identified in *Diploneis sp.* by the LC-MS analysis, only trace amounts of LOX products are reported in this species. 

### 2.2. FOX2 Assay

Fatty acid hydroperoxides (FAHs) are the primary products of LOX enzymes and their quantitation with the FOX2 assay allows inferring the activity of LOX enzymes in diatoms [35,39]. In good agreement with the amount of oxylipins detected by LC-MS, analysis showed significantly different levels of FAHs (*p*-value < 0.001) in the four species. As shown in Figure 2b, the highest concentration of FAHs was found in *N. shiloi* (1.4 ± 0.05 μmol of FAHs per mg C) while the lowest level in *C. scutellum* (0.28 ± 0.01 μmol mg^−1^ C; *p*-value = 0.0012). 

In addition, FOX2 activity (0.016 ± 0.001 μmol FAHs per mg C; *p*-value < 0.001) was at the limit of the detectable threshold in *Diploneis* sp. that is free of lipoxygenase products. Overall, LOX activity linearly correlates with oxylipin levels quantified in the four benthic species (R = 0.97, *p*-value < 0.001, see Figure 2c). This result supports the biosynthesis of LOFAs by oxidative pathways not dependent on lipoxygenase enzymes in *Diploines*. 

### 2.3. Identification of Lipoxygenase Pathways of C. closterium

As shown in Figure 3, MS/MS fragmentation of the epoxyalcohol of *C. closterium* around at 8.42 min (*m*/*z* 371) produced three fragments at *m*/*z* 259, 273 and 289 in agreement with the occurrence of 14,15-epoxy-13-hydroxy-eicosa-5Z,8Z,11Z,17Z-tetraenoic acid (14,13-EHETE) [26,34]. Analogously, fragmentation of the product at 9.87 min (*m*/*z* 397) gave a cluster of three fragments at 285, 299 and 315 that are diagnostic of 16,17-epoxy-15-hydroxy-docosa-4*Z*,7*Z*,10*Z*,13*Z*,19*Z*-pentaenoic acid (16,15-EHDPE) [38]. The identification of these LOFAs established lipoxygenation at C-15 of EPA and at C-17 of DHA, in agreement with synthesis of 0.0045 ± 0.0006 μmol/mg of C of 14,13-EHETE, 0.0027 ± 0.0001 μmol/mg of C of 15-hydroxy-eicosa-5Z,8Z,11Z,13E,17Z-pentaenoic acid (15-HEPE,) and 0.0013 ± 0.0001 μmol/mg of C of 16,15-EHDPE.

### 2.4. Identification of Lipoxygenase Pathway of C. scutellum

MS/MS analysis of the product at 8.48 min (*m*/*z* 371) in the LC-MS profile of *C. scutellum* was consistent with the fragments of 14,13-EHETE (MS cluster at *m*/*z* 259, 273 and 289) derived by EPA-dependent 15-LOX (Figure 3). On the basis of this assignment, the oxylipins of this diatom were quantified in 0.0012 ± 0.0001 μmol mg^−1^ C of 14,13-EHETE and 0.0011 ± 0.0007 μmol mg^−1^ C of 15-HEPE.

### 2.5. Identification of Lipoxygenase Pathway of N. shiloi

The extracts of *N. shiloi* featured the methyl esters of two major epoxyalcohol fatty acids corresponding to derivatives of HTrA (*m*/*z* 319, C_17_O_4_H_28_ [M + Na]^+^) and EPA (*m*/*z* 371, C_21_O_4_H_32_ [M + Na]^+^). As shown in Figure 3, MS/MS analysis of the first product gave a major fragment at *m*/*z* 221 and 207 according to the occurrence of 9,10-epoxy-11-hydroxy-hexadecadienoic acid (9,11-EHHDE, 0.002 ± 0.0007 μmol mg^−1^ C) produced by a HTrA-dependent 9-lipoxygenase previously described in the planktonic diatom *Skeletonema marinoi* [39]. MS/MS fragmentation of the second epoxyalcohol gave a single ion at *m*/*z* 233 in agreement with the occurrence of 11,12-epoxy-10-hydroxy-eicosa-5*Z*,8*Z*,14*Z*,17*Z*-tetraenoic acid (11,10-EHETE, 0.005 ± 0.0005 μmol mg^−1^ C) [26,34]. Synthesis of this oxylipin is due to EPA lipoxygenation by 11-LOX activity (Figure 3) that is also responsible for the presence of 0.0017 ± 0.0001 μmol mg^−1^ C of 11-hydroxy-eicosa-5*Z*,8*Z*,12*E*,14*Z*,17*Z*-pentaenoic acid (11-HEPE) in this diatom. 

### 2.6. GC-MS Analysis of PUAs 

*N. shiloi* was the only species among those under analysis that showed the presence of PUAs (Appendix A). GC-MS analysis indicated 2*E*,4*Z*-octadienal, 2*E*,4*Z*,7*Z*-decatrienal and 2*E*,4*Z*-decadienal, with 2*E*,4*Z*,7*Z*-decatrienal as the most abundant product. In agreement with previous reports with the planktonic diatoms *Thalassiosira rotula* and *Skeletonema marinoi* (formerly reported as *Skeletonema costatum*) [8,9], octadienal and decatrienal arise by the activity of two distinct hyperoxide-lyases (HPLs) that convert the hydroperoxides from peroxidation of HTrA by 9-LOX or EPA by 11-LOX, respectively. Despite the original report from Miralto et al. [6], decadienal is not usual in diatoms but has been reported from time to time both in laboratory strains and field samples [6,22,40]. Decadienal probably is the downstream product of lipoxygenation of arachidonic acid (AA, C20:4 ω-6) by the EPA-dependent 11-LOX that is also responsible for the synthesis of decatrienal [8,9,41]. Fatty acid analysis (data not shown) of *N. shiloi* revealed the presence of AA that supports the detection of decadienal in this species.

### 2.7. Effect of LOX Products on Sea Urchin Development

The quantities of FAHs and oxylipins measured through LC-MS analyses were correlated with biological assays of feeding experiments with the sea urchin *P. lividus* [31,32]. Specifically, the percentage of abnormal plutei deriving from adult sea urchins fed for one month with the four benthic diatoms (Appendix A) were analyzed together with the amount of oxylipins and FAHs found in the present study. With the hypothesis that detrimental effect on plutei was dependent on LOX activities, a ternary diagram of the three parameters (normalized oxylipin concentration, normalized FAH concentration, percentage of abnormal plutei) clearly showed a good clusterization of the data according to the biological species (Figure 4). 

The diatoms (*C. closterium*, *C. scutellum* and *N. shiloi*) with active LOX pathways, as deduced by synthesis of oxylipins and FAHs, formed a central group with secondary differences among the single species. On the other hand, *Diploneis* sp. was represented by an outgroup in agreement with the biochemical results. As shown in Figure 5a,b, the negative effect on sea urchin embryo development well correlated with the levels of oxylipins (*p*-value = 0.00074) and FAHs (*p*-value = 0.0043) in the case of the three species (*C. closterium*, *C. scutellum* and *N. shiloi*) with LOX products, whereas the statistical significance was weaker when *Diploneis* sp. was included in the analysis (Figure 5c,d; *p*-value = 0.012 and *p*-value = 0.031, respectively). 

## 3. Discussion

Lipoxygenase (LOX) pathways are common in marine diatoms but their occurrence was never investigated in benthic species. In this study, biochemical and MS analysis of four benthic diatoms clearly reveals LOX signatures in three of them, namely *C. closterium*, *N. shiloi* and *C. scutellum*. Only *Diploines* sp. apparently lacks this metabolism. The data support species-specific synthesis of poly-unsaturated aldehydes (PUAs) and linear oxygenated fatty acid (LOFAs) with overall levels of oxylipins that are in agreement with LOX-dependent synthesis of FAHs from polyunsaturated fatty acids including EPA (C20:5), AA (C20:4), DHA (C22:6) and HTrA (C16:3 ω-4) (Appendix A). The variety and type of these oxylipins suggests that there are no differences with LOX pathways previously described in other diatoms (Figure 6). 

In fact, major products of *C. closterium* and *C. scutellum* derive from EPA by 15-LOX. This metabolism is common to many diatoms [23,26,34,36] and the resulting oxylipins were linked to the toxic activity on egg production and hatching success of the copepod *Temora stylifera* [42]. The enzyme prefers EPA as a primary substrate but it is possible that it can recognize also DHA with addition of molecular oxygen at C17 [38]. The simultaneous presence of products deriving by 15-LOX metabolism of EPA and DHA is a biochemical characteristic that *C. closterium* shares with the planktonic diatoms of the genus *Leptocynlindrus* [38]. However, the most striking analogies between planktonic and benthic diatoms are found between *N. shiloi* with *S. marinoi* and *T. rotula*. In these three species, the mixture of oxylipins derives from activation of both C_16_ polyunsaturated fatty acids by 9-LOX and EPA by 11-LOX [7,8,9,10,17,18,24,37,39,43,44]. These pathways are responsible for the synthesis of LOFAs and PUAs including decadienal from arachidonic acid (AA, C20:4 ω6). In addition, *N. shiloi* produces a considerable amount of oxygenated derivatives of palmitoleic acid, namely 9-KHME and 9-HHME, that were so far only described in *T. rotula* and *S. marinoi* [25,39]. It is noteworthy that the whole array of these oxylipins was reported in mesocosm samples of marine plankton [37,43] and was correlated with the detrimental effects of diatoms on grazing copepods [6,12,14,20,39,45,46,47,48,49].

In agreement with the detrimental role of diatom LOX on marine grazers, survival of sea urchins plutei inversely correlate with levels of oxylipins and FAHs in *C. closterium*, *N. shiloi*, *C. scutellum*. By using the three best fits for 17 feeding experiments (Appendix A) [31,32], these species cluster in separate groups that are fully consistent with their negative effects on sea urchin embryos after diatom ingestion (Figure 4). In general, the higher the quantity of oxylipins and FAHs in the benthic species, the higher the number of abnormal plutei spawned (Figure 5a,b). In agreement with the results on planktonic copepods [46,50], the detrimental response of sea urchin embryos seems to be due to an accumulation of LOX-derived metabolites and does not depend on a specific class of oxylipins. 

The only apparent divergence from this scenario is *Diploneis* sp. that, despite the absence of detectable levels of oxylipins and LOX activity (Figure 2), induces aberrations on embryo development of sea urchins [32]. In the cluster analysis this species forms an outgroup whose inclusion in the statistical analysis reduces the correlation between plutei survival and levels of oxylipins and FAHs. The LC-MS profile of *Diploneis* sp. includes a family of undetermined metabolites that share more than one chemical characteristic with oxygenated derivatives of fatty acids. The occurrence of other mechanisms of lipid oxidation in diatoms is indirectly supported by occasional reports of unusual fatty acid derivatives like bacillariolides [51,52]; thus it is conceivable that the products of *Diploneis* sp. may derive from oxidative pathways independent of lipoxygenase activity. It is possible that these still uncharacterized mechanisms of lipid peroxidation in *Diploneis* sp. may have the same biological role of LOX pathways in the other species. Characterization of these compounds and the related enzymes is in progress.

## 4. Materials and Methods

### 4.1. General

Carbon, hydrogen, nitrogen, and sulfur (CHNS) were analyzed on a Thermo Electron Corporation FlashEA 1112 CHNS elemental analyzer with autosampler (Thermo Fisher Scientific, Waltham, MA, USA) [53]. Gas chromatography-mass spectrometry (GC-MS) was carried out by a Polaris Q Ion Trap (Thermo Fisher Scientific, Waltham, MA, USA). Liquid chromatography-mass spectrometry (LC-MS) was performed by a Micro-qToF mass spectrometer (Waters) with electrospray ionization (ESI) source (positive mode) and coupled with a Waters Alliance HPLC system equipped with a C-18 Kromasil column (4.6 × 250 mm, 100 Å, Phenomenex) and by a Q-Exactive Hybrid Quadrupole-Orbitrap spectrometer (Thermo Fisher Scientific, Waltham, MA, USA) with an ESI source and interfaced to an Infinity 1290 UHPLC System (Agilent Technologies) equipped with a Luna 5μ C-18 column (2.0 × 150 mm, 100 Å, Phenomenex). Samples were dried by a Buchi R-200 Rotavapor (Marshall Scientific, Hampton, NH, USA). 

### 4.2. Cell Culturing and Collection

Monoclonal cultures were selected from diatoms previously isolated from *Posidonia oceanica* leaves collected in Lacco Ameno d’Ischia (Bay of Naples, Italy: 40°44′56″ N, 13°53′13″ E). They were maintained in 12 well-multiwell plates in *f*/*2* medium (Sigma Guillard’s) in a thermostatic chamber at 18 ± 1 °C and 140 μmol (photons) m^−2^ s^−1^, with a 12:12 h (light:dark) photoperiod. Massive cultures were prepared by inoculating the same strains in 17-cm glass Petri dishes [54] filled with f/2 medium until the stationary phase, normally reached in about 15 days of culture, under the same conditions described above. Cells were detached from the glass using a blade and then collected by a sterile Pasteur pipette and transferred in a flask prior to centrifugation at 3000× *g* for 10 min at 4 °C, then frozen in liquid nitrogen and kept at −80 °C until analysis. 

### 4.3. Elemental Analysis by CHNS Analyzer

About 300 mg wet biomass of each diatom stored at −80 °C was firstly dried overnight using a freeze-drying chamber and then disrupted into small pieces using a pestle. Each species (~3 mg dry weight) was weighted in a tin sample cup with vanadium pentoxide (catalyst). Sulfanilamide was used as control. Samples were analyzed in triplicates.

### 4.4. GC-MS Analysis of Total Fatty Acids

Fatty acid composition was determined by methanol extraction on 300 mg wet biomass. Pellet suspension in methanol (5 mL/g pellet) was sonicated (5 min at 37 kHz) prior to centrifugation at 3000× *g* for 6 min at 4 °C. The supernatant was recovered and the residue was extracted another two times using the same procedure. Supernatants were combined and dried. Part of the oily residue (0.5 mg) was saponified by catalytic amount of Na_2_CO_3_ in 500 μL MeOH at room temperature overnight under magnetic stirring. After neutralization by 1% acetic acid, the methyl esters were extracted by partition between diethyl ether and water. The ether layer was dried under nitrogen and the resulting material was dissolved in *n*-hexane (1 μg μL^−1^) and analyzed by GC-MS (70 eV) on a 5% diphenyl column (inlet temperature of 270 °C, transfer line set at 280 °C and ion source temperature of 250 °C). Elution of fatty acid methyl esters (FAMEs) was accomplished by an increasing gradient of temperature according to the following program: 200 °C for 2.5 min then 15 °C/min up to 290 °C, followed by 7 min at 290 °C. Each sample (2 μL) was directly injected. Fatty acids were identified by comparing retention time (RT) of peaks to standard fatty acid methyl asters (PUFA-1, Marine Source, Sigma-Aldrich).

### 4.5. LOX Activation and Oxylipin Extraction

Oxylipins were extracted and analyzed according to our previous methods [8,26,34]. Briefly, 4*E*-decenal (60 μg/g of pellet) and 16-hydroxy-hexadecanoic acid (10 μg/g of pellet) were added to 1 g diatom cell pellet (wet weight). Cells were suspended in 1 mL of 0.5 M NaCl in 50 mM TRIS HCl at pH 8 and sonicated for 5 min. Sample was left at room temperature for 30 min prior to addition of 1 mL acetone. The resulting green homogenate was centrifuged at 3000× *g* for 6 min at 4 °C. Supernatant was recovered and pellet was extracted in H_2_O-acetone 1:1 another two times. Supernatants were combined and partitioned against an equal volume of CH_2_Cl_2_ twice. The combined organic phases were dried over Na_2_SO_4_ and evaporated under reduced pressure by a rotary evaporator. 

### 4.6. Analysis of Polyunsaturated Aldehydes (PUAs) and Linear Oxygenated Fatty Acids (LOFAs) 

PUAs were determined as carbetoxyethylidene (CET)-triphenylphosphorane derivative (1.2 mg per mg of extract) by GC-MS [8]. For LOFAs, crude diatom extracts were methylated by diazomethane (0.4 mL per 10 mg extract) for 30 min at room temperature. After removing the organic solvent under nitrogen, the residues were recovered in MeOH and analyzed by two different LC-MS platforms based on either Time-of-Flight (TOF) or Orbitarp analyzers according to our standard protocols [26,34]. Compounds were quantified by Xcalibur^™^ software and values were normalized by carbon content as measured in the same samples by elemental analysis.

### 4.7. Assessment of LOX Activity 

LOX activity was established according to colorimetric measurement of fatty acid hydroperoxides (FAHs) by modified ferrous oxidation-xylenol orange 2 (FOX2) assay [33]. Lipoxygenases were activated as described above by lysis of 50 mg diatom pellets in 500 μL of 0.5 M NaCl in 50 mM Tris HCl at pH 8. Aliquot of the resulting homogenates were diluted to 200 μL by the same buffer solution and added to 800 μL freshly prepared FOX2 reagent. Samples were vortexed and left at room temperature for 15 min prior to centrifugation at 11,290× *g* for 4 min at 16 °C. A parallel preparation was carried out by adding tris-(2-carboxyethyl)-phosphine (TCEP) prior to FOX2. A buffer solution with FOX2 reagent was used as control. Lipoxygenase activity was expressed as difference between absorbance at 560 nm of samples without and with TCEP treatment. FAHs concentrations were measured using a standard curve of increasing concentrations of H_2_O_2_. Measurements are duplicates of three scalar concentrations of cell homogenates. Final values were normalized for carbon content, as measured by elemental analysis. 

### 4.8. Feeding Experiments on the Sea Urchin Paracentrotus lividus

The selected diatom strains were produced in axenic conditions in thermostatic chambers and further collected and incorporated into a 2% agar matrix to produce a diatom enriched feed simulating leaves of *P. oceanica* with diatom epiphytes that represent a common food for sea urchins. Sea urchins were fed ad libitum [31,32] and feed residuals were removed every day along with sea urchin’s fecal pellets. After 30 days, the treated animals were sacrificed and their gametes were collected to be used for in vitro fertilization. The rates of fertilization, as well as first cleavage and the percentage of embryo and larvae abnormalities were recorded. Data were elaborated by statistical methods as described below.

### 4.9. Statistical Analyses 

Data collected for each parameter (Oxylipins, FAHs and Percentage abnormal plutei) were measured on three biological replicates of the four benthic species (*C. closterium*, *N. shiloi* and *C. scutellum* and *Diploneis* sp.) and reported as mean ± standard deviation (M ± SD). Statistical significance of differences among four benthic species was evaluated using One-Way ANOVA test (a parametric test) for Oxylipins and FAHs. Kruskal-Wallis rank sum test (a non-parametric test which was used when ANOVA assumptions are not verified) was applied for abnormal plutei (%). To test that the data are normally distributed and the variance across species are homogeneous we used Shapiro-Wilk test on the ANOVA residuals (i.e., if the *p*-value > 0.05, then the normality is not violated) and Levene’s test (i.e., if the *p*-value > 0.05, then we assume the homogeneity of variances in the different groups), respectively. As the statistical test is significant (*p*-value < 0.05), a post-hoc analysis based on Tukey multiple pairwise-comparisons was applied to determine significance of mean difference between specific pairs of species (signif. codes: 0 ‘***’, 0.001 ‘**’, 0.01 ‘*’, 0.05 ‘.’, 0.1 ‘ ’, 1). Parameters (Oxylipins, FAHs and Percentage of abnormal plutei) were clustered in a ternary diagram. Correlation analysis was performed for each pairwise combination (Oxylipins-FAHs; oxylipins-abnormal plutei (%); FAHs-Abnormal P lutei (%)) with Pearson correlation coefficient (*R*) the corresponding *p*-value (*p*). Correlations tests (significant *p*-value below 0.05) were performed in presence or absence of *Diploneis* sp. In correlation and clustering analysis, data were normalized by the highest value within each species. To have the same size (*n* = 3) of each parameter, the three best fits of 17 feeding experiments for each diatom species were used for the clustering and correlation analyses of abnormal plutei. Best fittings were defined as values with the smallest *p*-value by One-Way ANOVA test for all the possible 3-permutation of the 17 replicates. Analysis was carried out by R software environment for statistical computing and graphics (*ggplot2* and *ggpurb* R packages for correlations analysis; *ggtern* R package for the ternary diagram). 

## 5. Conclusions

Lipid peroxidation generates specific classes of ROS, namely peroxyl and alkoxy radicals, and is involved in numerous physiological and pathological states across biological kingdoms. We already showed that the teratogenic activity of planktonic diatoms can be linearly related to the onset of ROS (reactive oxygen species) activity after fatty acid peroxidation [39]. In planktonic diatoms species that possess strong LOX activity, this process is mostly dependent on LOXs, and synthesis of oxylipins and FAHs strongly correlates with detrimental effects on grazers such as reduced hatching success and spawning of teratogenic offspring. The present study is the first report of LOX pathways leading to the production of oxylipins, both PUAs and LOFAs, in benthic diatoms. In analogy with planktonic species, the occurrence of these products correlates with the detrimental effects due to ingestion of diatoms by sea urchins, thus suggesting a possible correlation with the complex network of interactions that sustain formation and development of benthic associations [30]. The biological activity of *Diploneis* sp. on the sea urchin *P. lividus* suggests that biochemical pathways independent of LOXs can also be related to these mechanisms.

## Figures and Tables

**Figure 1 marinedrugs-18-00066-f001:**
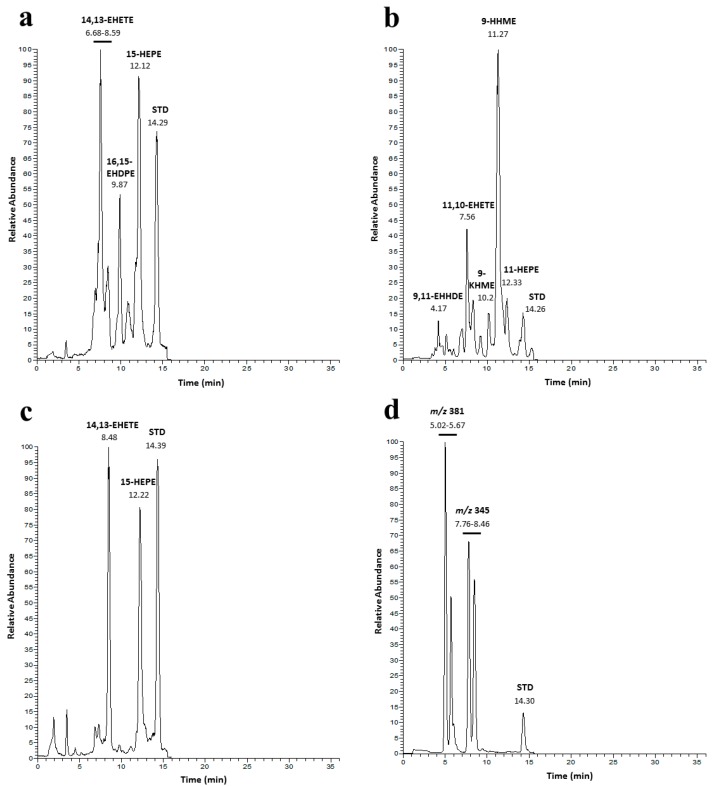
LC-MS analysis of the four benthic diatoms. Chromatographic profiles of (**a**) *C. closterium*, (**b**) *N. shiloi*, (**c**) *C. scutellum* and (**d**) *Diploneis* sp.; Internal standard: 16-hydroxy-hexadecanoic acid (**STD**). Abbreviations are in agreement with d’Ippolito et al. [34]. Structures are reported in Appendix A.

**Figure 2 marinedrugs-18-00066-f002:**
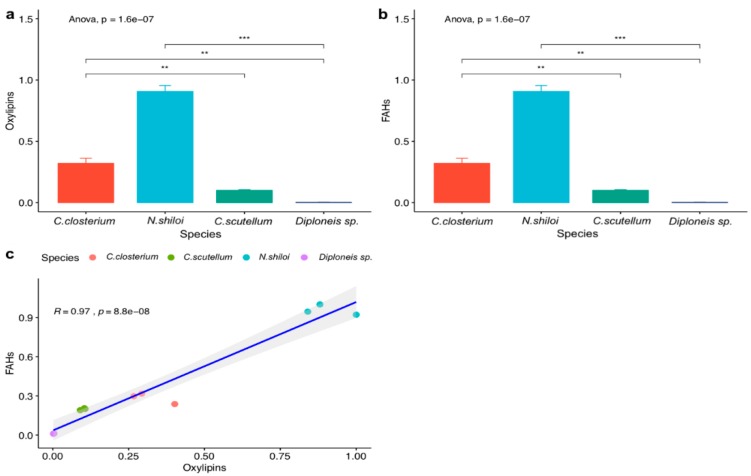
LOX activity in the four benthic diatoms. (**a**) Levels of oxylipins as μmol per carbon mg; (**b**) Levels of FAHs as μmol per carbon mg; (**c**) statistical correlation between oxylipin and FAHs levels. Data are reported as means ± SD (*n* = 3). Statistical differences were evaluated by One-Way ANOVA followed by Tukey’s post-hoc analysis for multiple comparisons (Appendix A; * *p* < 0.05, ** *p* < 0.01, *** *p* < 0.001).

**Figure 3 marinedrugs-18-00066-f003:**
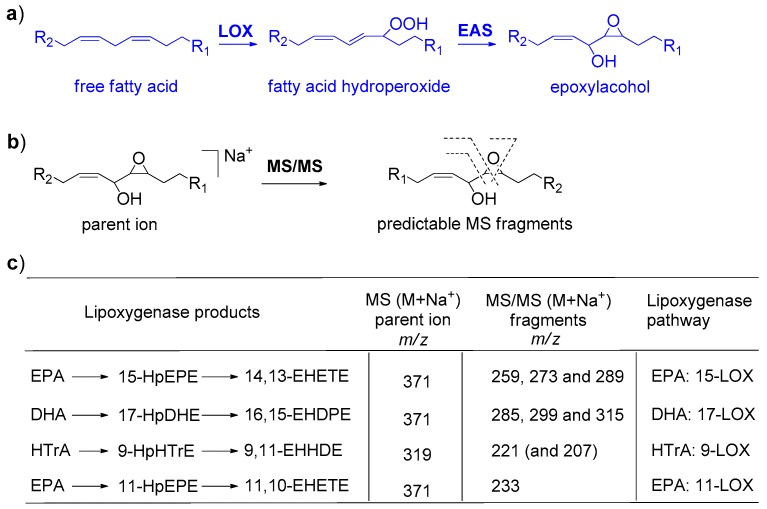
Identification of active lipoxygenases by MS/MS fragmentation of epoxyalcohols in *C. closterium*, *N. shiloi* and *C. scutellum*. For simplicity, the polyunsaturated chain of EPA is represented by the 1,3-pentadiene moiety that undergoes to enzymatic oxidation. R_1_ and R_2_ are variable alkyl residues to complement the structure of EPA (C20:5) or HTrA (C16:3). (**a**) Biosynthesis of fatty acids by LOX pathways; (**b**) MS parent ion and MS/MS fragmentation of epoxyalcohols analyzed as methyl esters in positive mode; (**c**) Diagnostic MS fragments used for the regiochemical identification of epoxyalcohols of diatoms. Assignments and abbreviations are in agreement with d’Ippolito et al. [34]. Structures are reported in Appendix A.

**Figure 4 marinedrugs-18-00066-f004:**
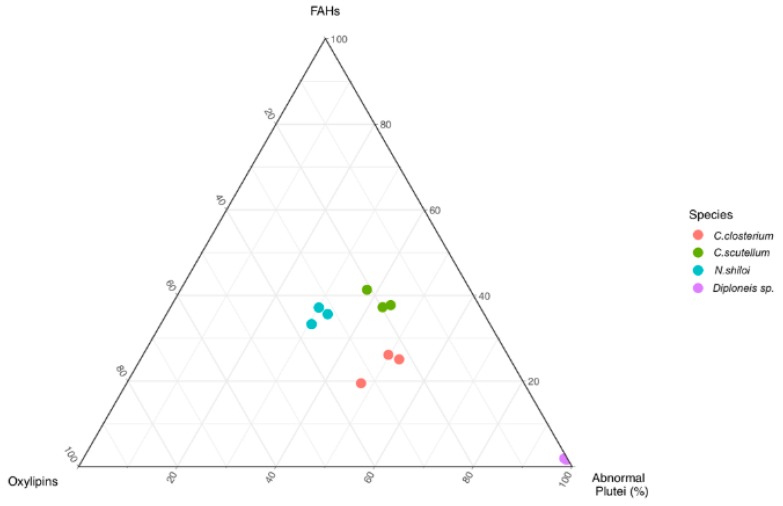
Ternary diagram based on survival of sea urchin plutei, level of oxylipins and level of FAHs. Graphical representation derives by sum to unity (100%) of the respective variables with percentage of abnormal plutei that is strictly dependent on oxylipins and FAHs levels. Single values are reported in Appendix A.

**Figure 5 marinedrugs-18-00066-f005:**
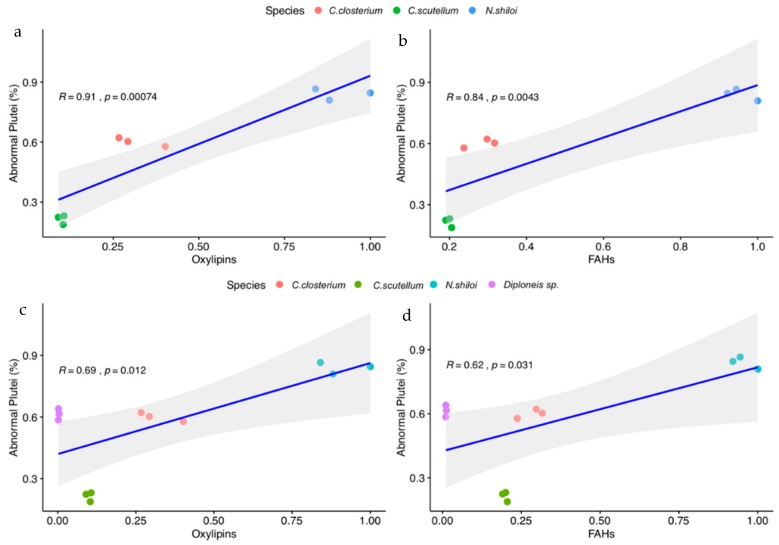
Correlation between feeding effect on the sea urchin *Paracentrotus lividus* and LOX products in the four benthic diatoms. (**a**,**b**) correlation of abnormal plutei with oxylipins and FAHs in the three species with active LOX pathways (*C. closterium, N. shiloi and C. scutellum*); (**c**,**d**) correlation of abnormal plutei with oxylipins and FAHs including *Diploneis* sp.

**Figure 6 marinedrugs-18-00066-f006:**
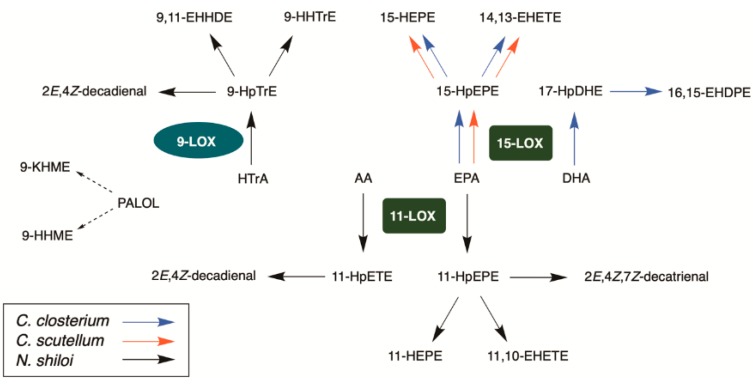
Lipoxygenase pathways and oxylipins of *C. closterium*, *N. shiloi* and *C. scutellum* (see abbreviation section for the name of the compounds). Abbreviations are in agreement with d’Ippolito et al. [34]. Structures are reported in Appendix A.

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
