# Peer review of "Lipoxygenase Pathways in Diatoms: Occurrence and Correlation with Grazer Toxicity in Four Benthic Species"

_marinedrugs, 2020, doi:10.3390/md18010066_

Round 1

Reviewer 1 Report

In their work, Ruocco et al investigated the lipoxygenase pathways in diatoms by applying biochemical and GC-MS analyses. Authors have shown the LOX pathways in C. closterium, N shiloi and C. scutellum whereas Diploines species, though produced other fatty acid derivatives, displayed lack of LOX metabolism PUAs and LOFAs.

Overall, the paper is well written and experiments are well done to study correlation of biochemical pathways in the production of oxylipins such as PUAs and LOFAs in benthic and planktonic diatoms.

Reviewer has a concern over the way LOX activation and the assessment of LOX activity was done in four banthic diatoms. PLease explain in detail.

Second concern in the why only four benthic diatoms were selected for the study?

Author Response

Reviewer #1

In their work, Ruocco et al investigated the lipoxygenase pathways in diatoms by applying biochemical and GC-MS analyses. Authors have shown the LOX pathways in C. closterium, N shiloi and C. scutellum whereas Diploines species, though produced other fatty acid derivatives, displayed lack of LOX metabolism PUAs and LOFAs.

Overall, the paper is well written and experiments are well done to study correlation of biochemical pathways in the production of oxylipins such as PUAs and LOFAs in benthic and planktonic diatoms.

Authors reply

We thank the reviewer for the appreciation of our work and for the careful evaluation of our manuscript.

Reviewer #1

Reviewer has a concern over the way LOX activation and the assessment of LOX activity was done in four banthic diatoms. Please explain in detail.

Authors reply

LOX activation was triggered by sonication and oxylipins were identified as methyl ester  by LC-MS method. Both these processes are widely detailed in the cited references. However, we have strongly improved the description of the methods in Figure 3 and slightly detailed the method in the discussion.

Reviewer #1

Second concern in the why only four benthic diatoms were selected for the study?

Authors reply

We selected these four benthic diatoms for this study because only for these species we have the correlation with grazer toxicity as reported in our papers Ruocco et al. (2018) e Ruocco et al. (2019)

Reviewer 2 Report

The manuscript submitted  by Ruocco and co-workers summarises the main results about the role of oxylipins (LOFAs) produced by for species of benthic diatoms.

The topic is of mid soundness and fits to the scope of this journal. The manuscript is in general well written and organised. The use of English (typos and grammar) must be revised and some key details are missing in materials an methods section in order to make the experiments reproducible.

Comments have been embedded through the manuscript in order to help the authors to improve this version.

Author Response

Reviewer #2

The manuscript submitted by Ruocco and co-workers summarises the main results about the role of oxylipins (LOFAs) produced by species of benthic diatoms.

The topic is of mid soundness and fits the scope of this journal. The manuscript is in general well written and organized. The use of English (typos and grammar) must be revised and some key details are missing in the materials and methods section in order to make the experiments reproducible.

Authors' reply

We thank this reviewer for the careful evaluation of our manuscript and the suggested revisions that have improved the quality of this paper.

Reviewer #2

Comments have been embedded through the manuscript in order to help the authors to improve this version.

Authors' reply

Every comment of this reviewer has been addressed and every suggested correction has been made in the revised manuscript. A detailed list of corrections is reported in the manuscript with changes highlighted by the track-change feature.

Reviewer 3 Report

Please find my comments about the results and the presentation below:

There is no introduction in the Results chapter. In my opinion there should be a section describing the way the organisms were obtained, kept and cultured. Instead the authors described directly derivatives of fatty acids. Obtained data on fatty acids (MS, GC and other results) should be transferred to the Methods chapter. Figure 3: This schema is just too small. Please make it large enough to be easily readable. Line 166: there is: C. Closterium and should be: C. closterium In my opinion, the authors didn’t pay enough attention on discussion of the results obtained during the analysis of Diploneis Figure 6: This schema requires explanation of all abbreviations in the legend. Some relationships are written as abbreviations while others have full names, why? Please change the title of 4.3 chapter. Please add chapter Conclusions.

Author Response

Reviewer #3

There is no introduction in the Results chapter. In my opinion there should be a section describing the way the organisms were obtained, kept and cultured. Instead the authors described directly derivatives of fatty acids.

Authors' reply

Details about isolation, conservation and culturing of microalgae are reported in the Materials and Methods section. According to the reviewer's comment, we have added a brief introduction at the beginning of the Results in the revised version of our manuscript.

Reviewer #3

Obtained data on fatty acids (MS, GC and other results) should be transferred to the Methods chapter.

Authors' reply

This information was already present in the Methods section (see paragraph 4.4).

Reviewer #3

Figure 3: This schema is just too small. Please make it large enough to be easily readable.

Authors' reply

This figure has been improved in clarity and readability.

Reviewer #3

Line 166: there is: C. Closterium and should be: C. closterium

Authors' reply

This change has been made in the revised manuscript. 

Reviewer #3

In my opinion, the authors didn’t pay enough attention on discussion of the results obtained during the analysis of Diploneis

Authors' reply

We apologize but this comment is not clear to us. The methods and analytical work-out have been widely discussed in our previous papers. References to these papers are present in the bibliography of this new submission. Thus, we believe that it is not necessary to provide further details about the chemical methodology. On the other hand, we have carefully evaluated and discussed the biological and biochemical meaning of our results.

In the revised manuscript, we have provided further details (Figure 3 and the corresponding text) and improved the discussion. We hope that this improvements are sufficient for the reviewer. Otherwise we will be very happy to answer to further requests or specific comments. 

Reviewer #3

Figure 6: This schema requires explanation of all abbreviations in the legend. Some relationships are written as abbreviations while others have full names, why?

Authors' reply

The full name and abbreviation of each compound is reported in the list of abbreviations at the end of the manuscript. However, in agreement with the reviewer's comment, we made more homogeneous the use of names and abbreviations throughout the manuscript including the abbreviations used in this Figure. 

Reviewer #3

Please change the title of 4.3 chapter.

Authors' reply

This change has been made in the revised manuscript. 

Reviewer #3

Please add chapter Conclusions.

Authors' reply

the Conclusions section has been added in the revised manuscript.

Round 2

Reviewer 2 Report

Thank you very much for your time and effort addressing all the comments and suggestions made by this reviewer.

Reviewer 3 Report

I don't have more question